

# Assessing heat and freshwater changes in the Southern Ocean using satellite-derived steric height

Jennifer Cocks[1,2], Alessandro Silvano[3], Alice Marzocchi[1], Oana Dragomir[3], Noémie Schifano[4], Anna E. Hogg[2], Alberto C. Naveira Garabato[3]

[1]National Oceanography Centre, Southampton, SO14 3ZH, United Kingdom
[2]Institute for Climate and Atmospheric Science, University of Leeds, Leeds, LS2 9JT, United Kingdom
[3]School of Ocean and Earth Science, University of Southampton, SO14 3ZH, United Kingdom
[4]Univ. Brest, CNRS, IRD, Ifremer, Laboratoire d'Océanographie Physique et Spatiale, IUEM, Brest, France

*Correspondence to*: Jennifer Cocks (eejco@leeds.ac.uk)

**Abstract.** The Southern Ocean plays a central role in regulating the global overturing circulation, ventilating the deep ocean, and driving sea level rise by delivering heat to Antarctic ice shelves. Understanding heat and freshwater content in this region is key to monitoring these global processes and identifying multiyear changes; however, in-situ observations are limited, and often do not offer the spatial or temporal consistency needed to study long-term variability. Perturbations in steric height can reveal changes in oceanic heat and freshwater content inasmuch as they impact the density of the water column. Here, we show for the first time that the monthly steric height anomaly of the Southern Ocean south of 50°S can be assessed using altimetry and GRACE gravimetry data from 2002 to 2018. Steric height anomalies are validated against in-situ Argo float and CTD data from tagged elephant seals. We find good agreement in the ice-free ocean and parts of the seasonal ice zone, but that the uncertainty of steric height increases on the Antarctic continental shelf and within the permanent ice zone due to leakage error and anti-aliasing in GRACE. The open-ocean steric height anomalies exhibit spatio-temporally coherent patterns that: (i) capture the expected seasonal cycle of low (high) steric height in winter (summer); and (ii) reflect interannual anomalies in surface heat and freshwater content and wind forcing associated with positive and negative phases of the two major modes of Southern Hemisphere climate variability (the El Niño - Southern Oscillation and Southern Annular Mode).

## 1    Introduction

The water masses of the Southern Ocean are an essential component of the global thermohaline circulation, connecting the world's oceans and feeding ventilated surface waters into the abyssal ocean (Rintoul, S. 2018; Morrison et al., 2021). Recent changes in deep water and the overturning circulation (Lee et al., 2023; Gunn et al., 2023; van Wijk and Rintoul, 2014; Purkey et al., 2019) and unprecedented reductions in sea ice (Parkinson, 2019; Haumann et al., 2023; Purich and Doddridge, 2023) have stressed our need to understand the dynamics of the Southern Ocean, particularly those involving water mass transformations and heat and freshwater storage. The South Atlantic, Pacific and Indian Oceans feed into and circulate within the Southern Ocean, providing links to the global ocean via long-distance atmospheric and oceanic teleconnections (Armitage





et al., 2018; Turner, 2004; Hoskins and Karoly, 1981). As the global climate becomes increasingly extreme and unpredictable, it is essential to understand how the heat and freshwater content of the Southern Ocean is responding to climate drivers on multiyear timescales, in order to monitor and predict global changes in ocean circulation and deep-sea ventilation (Null et al., 2023).

The remoteness and hostility of the Southern Ocean has made it difficult to measure, and inter-model variability for key processes such as deep-water formation remains high (Heuzé, 2021). Despite improvements in the quality, frequency and distribution of oceanographic observations, coverage remains poor particularly in winter months and in permanently ice-covered areas (Smith et al., 2019). Improved coverage and quality of satellite data over the past decade has begun to fill this gap; we are now able to observe the sea surface height (SSH) of the Southern Ocean, including the ice-covered ocean (Armitage

et al., 2018; Auger et al., 2022), and more accurately measure sea ice concentration throughout the year (e.g. Eayrs et al., 2019; Parkinson, 2019; Kacimi and Kwok, 2020). Developments in gravimetry, specifically in the GRACE experiment, provide comprehensive observations of mass transports throughout the global ocean, with increasing accuracy within small ocean basins, near coastlines and in polar regions (Dobslaw et al., 2020; Dobslaw et al., 2017; Shihora et al., 2022). Here, we exploit these recent improvements in satellite technology to calculate steric height in the Southern Ocean, a measure related to water

column density, from which information on oceanic heat and salinity changes can be inferred.

Steric height is the contribution to SSH from changes in the density of the water column. Higher steric height values indicate a less dense (fresher and/or warmer) water column, while lower values denote a more dense (colder and/or more saline) water column. Steric height derived from satellite data agrees closely with in-situ observations in both global (Purkey et al., 2014; Feng and Zhong, 2015) and regional (Armitage et al., 2016; Karimi et al., 2022; Raj et al., 2020) studies, and can reveal large-

scale information about water column structure such as mixed layer depth (Gelderloos et al., 2013) and freshwater content (Lin et al., 2023; Armitage et al., 2016). Armitage et al. (2016) computed steric height for the Arctic, demonstrating the feasibility of the method in polar regions, which pose more of a challenge for satellite observations than lower-latitude oceans due to seasonally varying ice cover impacting altimetry returns (Bamber and Kwok, 2009; Tilling et al., 2018). The Southern Ocean presents a still more complex problem than the Arctic Ocean, due to strong winds and intense ocean currents driving

complex upwelling and downwelling patterns, and rough sea surface conditions that can affect the accuracy of satellite observations (Kacimi and Kwok, 2020; Kuo et al., 2008).

For clarity in this study, we divide the Southern Ocean into three basins: the Weddell-Enderby basin in the South Atlantic and Indian sectors, the Australian-Antarctic basin in the southern Indian ocean, and the Bellingshausen-Amundsen basin in the South Pacific (Fig. 1). The basins are further divided into seas, within which local steric processes can be more easily

constrained. The dynamics of each basin and sea are distinct from one another, and result in large regional seasonal and interannual differences in sea ice cover, and in heat and freshwater fluxes. Changes in these properties across the Southern Ocean are currently under scrutiny due to their impact on the global ocean and uncertain relationship to climate change. For example, the production of Antarctic Bottom Water (AABW) is being closely monitored due to its role as a driver for the overturning circulation and its apparent reduction over recent decades (Li et al., 2023; Meredith et al., 2011; Purkey and



Johnson, 2013), particularly in key formation regions like the Weddell Sea (Zhou et al., 2023) and the Australian-Antarctic Basin (Gunn et al., 2023). The deep waters of the Australian-Antarctic basin, which originate in the Ross Sea, have also freshened (Purkey and Johnson, 2013; van Wijk and Rintoul 2014; Rintoul, 2007; Purkey et al. 2019), while the Weddell Sea has shown signs of warming and freshening, partly due to a decrease in the denser classes of AABW (Meredith et al., 2011; Purkey and Johnson, 2013; Zhou et al., 2023; Strass et al., 2020). Ice sheet discharge has also been intensifying, particularly

in West Antarctica (Jacobs et al., 2022; Smith et al., 2020), and may also be having an impact on steric sea level (Rye et al., 2014; Brunnabend et al., 2015).

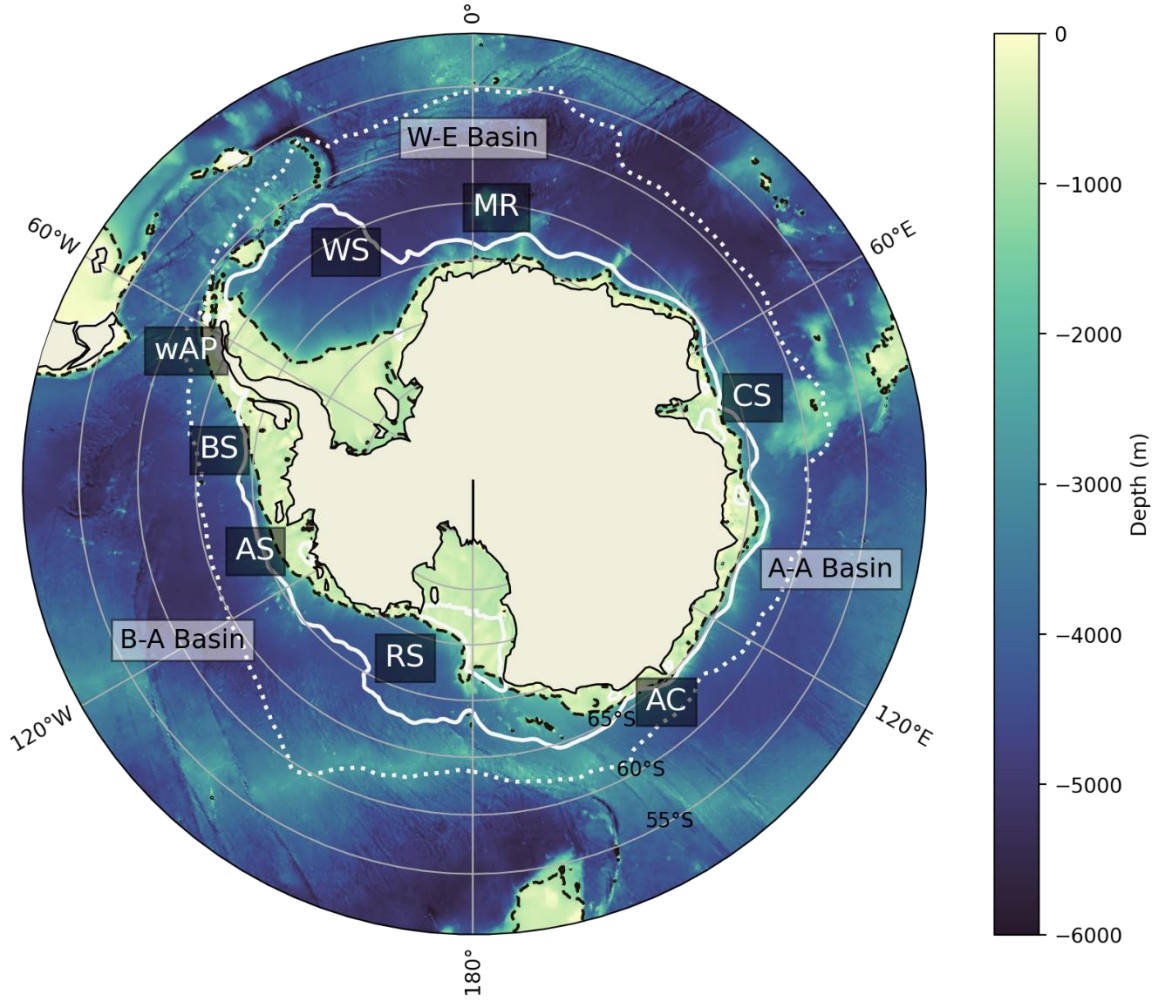

**Figure 1: Bathymetric map of the Southern Ocean showing the major ocean basins (Weddell-Enderby [W-E]; Australian-Antarctic [A-A]; Bellingshausen-Amundsen [B-A]) and the smaller regions of interest (Weddell Sea [WS]; Maud Rise [MR]; Co-operation Sea**
**[CS]; Adélie Coast [AC]; Ross Sea [RS]; Amundsen Sea [AS]; Bellingshausen Sea [BS]; western Antarctic Peninsula [wAP]). The extent of the seasonal ice zone (white, dotted) marks the area where the sea ice concentration (SIC) exceeds 0.15 on average in August, and the extent of the permanent ice zone (white, solid) marks where SIC is above 0.15 at least 95% of the time. Also marked is the -1000m isobath (black dashed).**





In this article, we use satellite altimetry and gravimetry from GRACE to reveal steric height anomalies south of 50°S during

2002-2018. We validate the steric height anomaly dataset against in-situ geopotential height computed from Argo floats   and

tagged elephant seal CTD data, and determine the regions for which this method is reliable and those within which additional

caveats apply. We consider both large and smaller areas to account for sampling inconsistencies in the in-situ data, and look

individually at the contributions from altimetry and GRACE to identify the source of uncertainty. We examine the steric height

results for the ice-free ocean and seasonal ice zone, discussing the multiyear time series and seasonal cycle and drawing links

to indices of the two major modes of Southern Hemisphere climate variability (the El Niño - Southern Oscillation and Southern

Annular Mode) to explain inter-annual fluctuations in steric height. We also give a more detailed analysis into the GRACE

contribution to steric height in Appendix B.

## 2    Data and Methods

### 2.1    Dynamic Ocean Topography

The Dynamic Ocean Topography (DOT) has been calculated using a pre-processed satellite altimetry product from the

combined Sea Surface Height (SSH) from Envisat (July 2002 to March 2012) and CryoSat-2 (April 2010 to July 2018)

(Dragomir, 2023). Along-track data from both satellite missions have been gridded onto a 1° longitude x 0.5° latitude grid,

and the data from CryoSat-2 and Envisat merged. The overlap between the satellites' operating periods has been used to

calculate and remove the offset between the two, which varies between 1-2 cm (Armitage et al., 2016). The resulting monthly

SSH has then been referenced against the GOCO05c geoid (Pail et al., 2016) to give DOT, and smoothed using a 300 km

Gaussian filter. We do not perform any further processing on the DOT.

### 2.2    Barystatic Height

We use the GRACE/GRACE-FO RL06 Mascon Solutions (version 2) from the Centre for Space Research (Save, 2020; Save

et al., 2016). The GRACE mascons represent gridded monthly means of the liquid water equivalent in metres of mass (i.e.,

barystatic) change relative to a baseline average from 2004-2009. Mascons were re-gridded onto the same grid as the DOT

using linear interpolation. Where GRACE data were missing, we linearly interpolated up to 2 months forward or back from

the last or next month with an available data point. Missing data more than 2 months apart from an available data point are left

as null values. There is a prominent data gap from July 2017 to May 2018 between the final recording from GRACE and the

first recording from GRACE-FO.

### 105    2.3    Steric Height

We combine the DOT and barystatic height (BH) data from GRACE to compute the steric height anomaly (SHA) of the

Southern Ocean (Fig. 2). Steric height represents the variation in SSH caused by density and can be used to infer changes in

water column buoyancy, stratification, or the mixed layer depth. It could also be used to examine how the relative quantities





of water masses of different densities are changing, or how the properties of a particular water mass are changing in regions
where the water mass in question dominates.

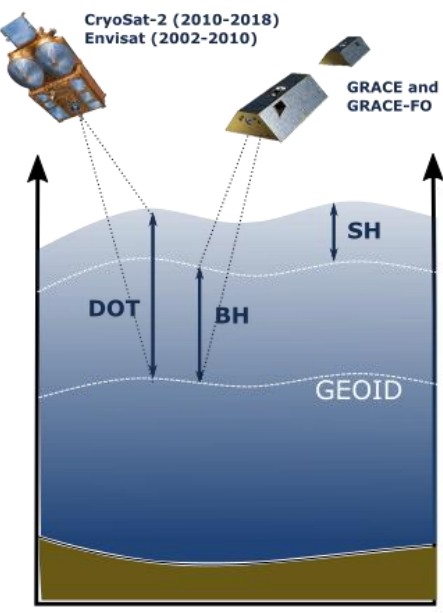

**Figure 2: Schematic showing the barystatic height (BH) and steric height (SH) contributing to the total dynamic ocean topography (DOT). BH represents the mass component is measured using the GRACE twin satellites, which evaluate gravitational field strength. DOT is the sea surface height relative to the geoid, and is measured by the altimeters on CryoSat-2 and Envisat. SH is the density component of DOT.**

We compute SHA by subtracting the BH anomaly (BHA) from the DOT anomaly (DOTA). This leaves only the density contribution, equivalent to SHA.

$$SHA = DOTA - BHA$$

We compute the DOT anomaly by removing the mean dynamic topography from July 2002 to June 2018. Since the time mean has been removed, the DOT anomaly is independent of the geoid and is equivalent to SSH anomaly.

The second term is equivalent to the GRACE mascon data, which is the water height equivalent of the ocean bottom pressure (Save et al., 2016), referred to as the BH in this manuscript. This is provided by CSR as the anomaly relative to a baseline of 2004 to 2009; however, we recompute the anomaly with respect to the baseline period July 2002 to June 2018 for consistency with the DOT data.

## 2.4    Geopotential Height Anomaly

We validate a sample of the derived SHA data against the geopotential height (GPH) computed from in-situ hydrographic profiles from Argo floats (Riser et al., 2018) and seal-mounted CTDs (Roquet et al., 2014).



We retain all profiles south of 50°S in which there are multiple pressure levels with a quality flag of 1 (indicating good quality data) for the adjusted temperature, salinity or pressure field. Depths where the quality of the adjusted temperature, salinity or pressure is anything but 1 are discarded. Profiles with a maximum depth less than 500 dbar or a minimum depth greater than 25 dbar are discarded. The maximum depth of 500 dbar is sufficiently deep to capture the largest differences between density profiles, but sufficiently shallow that enough profiles are retained to provide good spatial and temporal coverage. The upper limit of 25 dbar ensures that profiles collected by Argo floats trapped beneath ice are retained, in the case that they cannot reach the surface but continue to oscillate at depth. The locations of under-ice profiles are linearly interpolated between the points before and after the float became trapped. We do not omit under-ice profiles or correct their locations so that winter profiles are retained; however, this may result in loss of resolution, since measurements for a given grid cell may spread into adjacent cells.

The GPH is computed for each of the remaining profiles according to the equation below, where density $\rho$ has been calculated using the GSW density function (McDougall and Barker, 2011) using the adjusted pressure, temperature and salinity fields from the Argo floats and seal profiles, and reference density $\rho_{ref}$ is the density at 35 psu and 0°C for each depth level P in the adjusted pressure.

$$GPH = \int_0^{p_{500}} \frac{1}{\rho} - \frac{1}{\rho_{ref}} \, dp$$

$$\rho_{ref} = \rho(35psu, 0°C, P)$$

A large-scale comparison of the SHA and GPH data is performed by gridding the GPH data. We take the mean of the GPH from all profiles falling into the latitude, longitude and time bounds of each SHA grid point. This reduces the bias of increased profile density in certain regions and permits computation of a monthly grid-square anomaly of GPH (GPHA), which may be directly compared to the SHA values. However, missing data can skew the mean value for a given grid square towards seasons or years in which profiles are more abundant (i.e., summer), so consideration of the data frequency and quantity in each grid square should be given when using the gridded GPHA.

## 3    Results

### 3.1    Validation of SHA against in situ observations

We validate altimetry-derived SHA against the GPH computed from in-situ hydrographic profiles from Argo floats and seal mounted CTDs. GPH is the height of a water column of variable density relative to a reference density, and is analogous to steric height (e.g., Armitage et al., 2016). We first compute the gridded GPH (Methods) before calculating the GPH anomaly (GPHA) for each grid square, by subtracting the time-mean from each value. We use the mean GPH from July 2002 to June 2018 (Fig. 3a) to be consistent with the method for calculating the DOT and GRACE (thus, the steric height) anomalies.





**Figure 3: Validation of steric height anomaly (SHA) against geopotential height anomaly (GPHA) from Argo and seals profiles. (a) the mean Geopotential Height computed from profiles. (b) the Pearson correlation coefficient between the SHA and GPHA for each grid square. (c) the number of profiles recorded within each grid square. (d) the locations of recorded profiles within the region of highest data density (the area within the red box in (c)). (e) comparison between the SHA (turquoise) and GPHA (purple) timeseries averaged over the location shown in (d). Monthly measurements are shown with an 'x' and the 12-month rolling mean a solid line.**





We compute the Pearson correlation coefficient for the SHA and GPHA time series at each grid square (Fig. 3c), excluding pixels where data is available at fewer than 6 timestamps (Fig. 3b). The correlation between the SHA and GPHA is strongly positive across most of the Southern Ocean north of 60°S, and as far south as 65°S in the western Antarctic Peninsula (wAP) and the Australian-Antarctic basin (Fig. 3c). Furter south, a reliable comparison cannot be performed in most regions due to sparsity of in-situ data.

For a more detailed comparison of SHA and GPH we select an area where there are multiple in situ observations available each month, such that a reliable and long-term time series of GPHA can be constructed. We focus on the area around the Kerguelen Islands (65°E to 80°E, 55°S to 50°S), where there is a high density of profile data (Fig. 3d) due to this being the site of seal tagging and release (Roquet et al., 2014).

The 12-month rolling means of the monthly height anomalies from the satellite-derived SHA and profile-derived GPHA agree in both shape and magnitude, with both time series showing maxima in 2013 and 2017 and minima in late 2010 and 2015 (Fig. 3e). The magnitudes differ by up to 2 cm but are usually within 1 cm, demonstrating that this method is robust in the open Southern Ocean.

We use individual float data to validate the SHA within the seasonal ice zone, where the gridded GPHA dataset cannot provide a reliable comparison due to sampling sparsity, in Appendix A. we choose to focus our analysis on the open Southern Ocean, avoiding uncertain areas that are close to the coast, near to large ice shelves or glaciers, or subject to high-frequency mass transports, such as the coastal Ross and Weddell seas. We further minimise errors by considering changes only over wide time and space scales, such that seasonal or local errors will be smoothed out. We provide recommendations for progressing this method to better account for smaller and more coastal regions in Appendix B.



## 3.2 SHA in the ice-free ocean

**Figure 4: (a), (c), (e), (g) Timeseries of dynamic ocean topography anomaly (DOTA), barystatic height anomaly (BHA), steric height anomaly (SHA) and geopotential height anomaly (GPHA). Monthly anomalies for the Southern Ocean averaged outside of the seasonal ice zone (faint line) and the 12-month rolling mean (bold line). (b), (d), (f), (h) Seasonal cycle of DOTA, BHA, SHA and GPHA. Monthly anomalies for each year (faint lines) and climatology (bold line). Anomalies are relative to November 2002-October 2018 mean.**

We define the ice-free ocean by excluding the area where the sea ice concentration exceeds 0.15 in the month of maximum sea ice extent (August). Here, the seasonal cycle of the SHA closely matches that of the GPHA from the Argo and MEOP data in both shape and amplitude, and follows the expected seasonal variation, increasing in austral summer as the ocean warms and decreasing in the winter (Fig. 4f, h). Differences between the SHA and GPHA time series are to be expected due to sparsity of temporal, spatial and depth coverage of the Argo and MEOP data. The DOTA and BHA both show an increase of about 4 cm during 2002-2018 (Fig. 4a, c), leaving the SHA with no overall trend, once the two are combined. This suggests that, when averaged across the ice-free Southern Ocean, sea level rise is mostly due to mass increase and not steric effects. The 12-month rolling mean shows a slight decrease in SHA up until 2016, after which there is a sudden increase in 2017 followed by a



reduction in 2018. A similar pattern exists in the GPHA time series, though the two do not completely agree, possibly due to the patchiness of the in-situ data. There does not appear to be an overall trend in the data, which could reflect that length of 200 our time series is insufficient to capture density changes due to long-term heat uptake, e.g., relating to global warming, or that interannual variability is masking longer-term tendencies. In addition, long-term density changes are disproportionately distributed throughout the Southern Ocean such that there may be substantial cancellation between changes across different regions (Storto et al., 2019; Wang et al., 2017; Huguenin et al., 2022), resulting in a near-zero trend when a zonal average is taken.

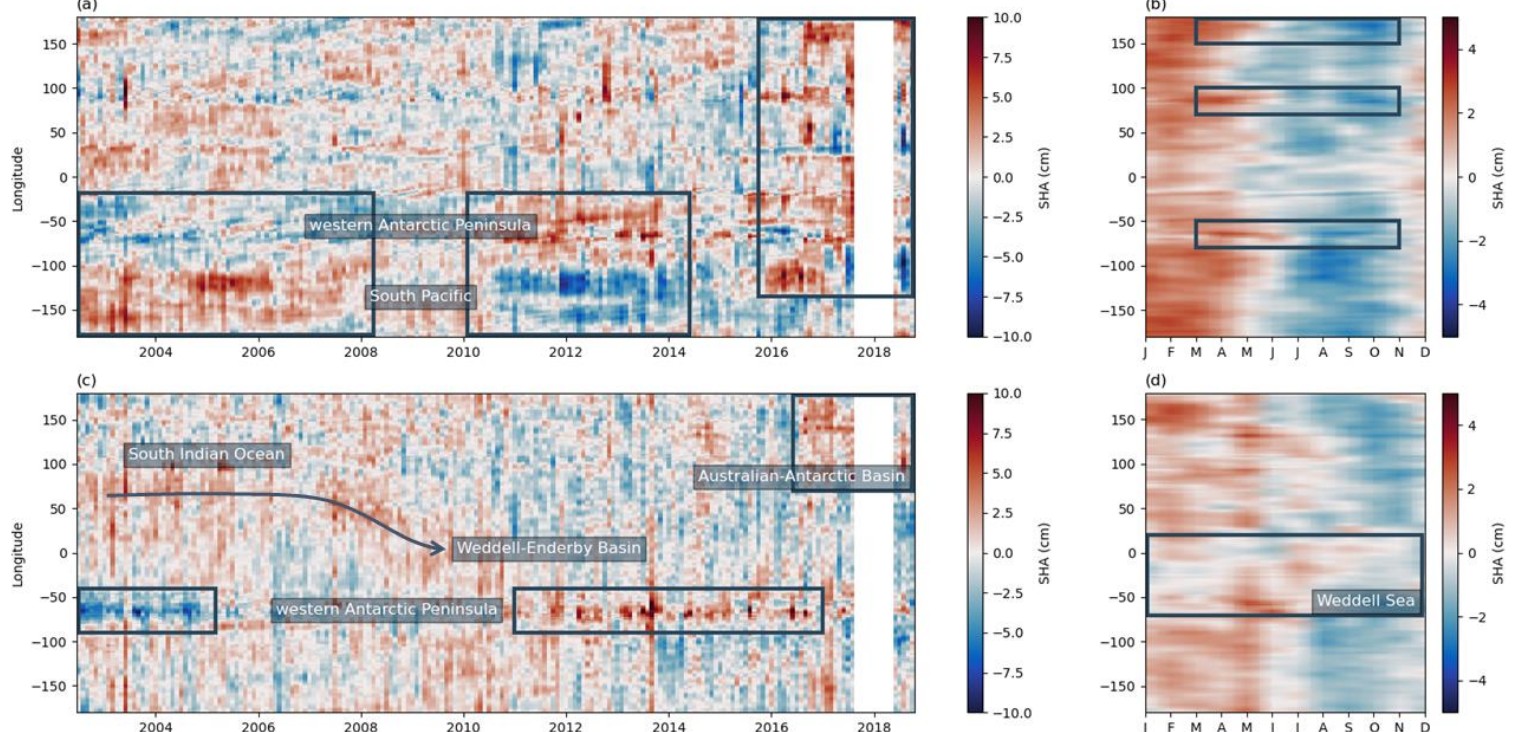

**Figure 5: a) Meridionally averaged steric height anomaly (SHA) with the seasonal mean (b) removed, for the ocean outside of the seasonal ice zone, and (c), (d) for the ocean in the seasonal ice zone. For clarity, points of interest described in the Results are shown in dark boxes. Points of interest in (a) include the dipole in the western Antarctic Peninsula (wAP) and the South Pacific and recent increases in SHA, particularly in the Australian-Antarctic (A-A) basin. In (b) are the delayed seasonal SHA maxima aligning with regions of higher topography. In (c) are the interannual change in the wAP, the recent increased SHA in the A-A basin and the** 210 **westward movement of higher SHA from the South Indian Ocean to the Weddell-Enderby (W-E) basin. In (d) is the delayed seasonal maximum in SHA in the Weddell Sea.**

Due to the regional differences in the ice and ocean dynamics of the Southern Ocean, we examine the spatial as well as temporal variability in the results. Changes in SHA outside the seasonal ice zone vary zonally, but show some coherency between certain regions (Fig. 5a). The SHA in the wAP is negative from 2002 to 2010, in contrast to the rest of the Bellingshausen-Amundsen 215 (B-A) basin, where SHA is positive, before a switch in 2011 after which the B-A basin displays strongly negative values until 2016, and the wAP positive. SHA values are generally most extreme in West Antarctica, with smaller anomalies in East





Antarctica, and everywhere after 2016 when strong positive anomalies begin to appear at more eastern locations and in the Australian-Antarctic (A-A) basin. Seasonally, there is little variation with longitude, except around 170°E, 80°E and 65°W, where the maxima and minima occur later in the year (Fig. 5b). These co-ordinates in the ice-free ocean correspond to island

locations, where the raised surrounding bathymetry and coastal effects could be delaying seasonal changes. It may also be that shallower topography introduces seasonal uncertainty, perhaps due to atmospheric or oceanic effects resulting in sudden changes in freshwater fluxes, which may cause errors in GRACE (Appendix B). However, based upon the comparison at the Kerguelen Islands, one example of where the delayed SHA maximum occurs, it does not appear that seasonal effects are impacting the quality of SHA results (Fig. 3e).

### 3.3  SHA in the seasonal ice zone

The area within the ice-covered ocean shows more variation than the ice-free ocean, due to changes in freshwater content driven by sea ice growth, movement and retreat, and ice shelf and glacial processes occurring along the Antarctic coastline. We divide the ice zone into the permanent ice zone (PIZ), computed by finding the grid squares whose sea ice concentration exceeds 0.15 (Vichi, 2022; Biddle and Swart, 2020) for more than 95% of the time, and the seasonal ice zone (SIZ), which is

the total ice zone minus the PIZ, the region where sea ice seasonally grows and melts.

The strongest SHA signal in the seasonal ice zone time series is in the wAP (Fig. 5c). Here, the steric height is anomalously low before 2006, then anomalously high during 2012-2017. The wAP exhibits strong freshwater content fluctuations due to high glacial discharge, precipitation and sea ice formation/melting (Meredith et al., 2021). Sea surface temperatures (SSTs) in this region are also heavily influenced by remote atmospheric and oceanic phenomena (Zhang and Duan, 2023; Li et al., 2021),

therefore it is likely that density changes here are driven by larger-scale climate variability. Indeed, the Southern Indian Ocean shows high SHA until 2006, when the positive signal appears to move towards the Prime Meridian in the Weddell-Enderby basin. This is followed by an increased SHA signal in the wAP. We also see strong positive anomalies in SHA after 2016, compared to relatively small and irregular fluctuations before this.

The seasonal cycle in the SIZ is weaker and more regionally variable than in the ice-free ocean (Fig. 5d). A late SHA maximum

occurs in May at all longitudes, and the ice-covered part of the Weddell Sea is particularly unusual, showing negative anomalies in January-March (summer). The SIZ encompasses the continental shelf, where shelf transports, strong katabatic winds and glacial discharge, as well as sea ice growth and formation, contribute to the freshwater budget. To better understand the role of sea ice, we compute the seasonal rate of change of SHA and sea ice concentration by differentiating each with respect to time in days (Fig. S5). The large-scale patterns show some consistency, with sea ice melt (indicated by a decreasing sea ice

concentration) associated with an increasing SHA, and sea ice growth with a decreasing SHA. There are some exceptions near the coast and in the Weddell and Ross seas; for instance, in austral autumn, when sea ice is forming most rapidly in the SIZ, the SHA is decreasing in most regions, but increasing in the Ross and Weddell seas. When sea ice is melting most rapidly in springtime (OND), SHA is decreasing in the Weddell Sea, Prydz Bay and close to the Ross Ice Shelf, but increasing everywhere else. The issues close to the coast and in the Weddell and Ross seas, where there are large ice shelves, are likely to have arisen





from the limitations described in Appendix A. Also, for a given location, we would not expect the change in sea ice concentration to be inversely related to SHA (as would be the case under purely thermodynamic effects) since dynamic effects, such as wind stress, cause spatial redistribution of surface freshwater fluxes from sea ice melt and growth (Holland and Kwok, 2012; Zhou et al., 2023).

## 3.4    SHA changes due to climate forcing

**Figure 6: a) Timeseries of Southern Oscillation (SO) Index smoothed over 12 months. Months showing an SO index greater than 1 standard deviation and less than -1 standard deviation highlighted in red. b) Composite plot of steric height anomaly (SHA) within months where the SO index is greater than 1 standard deviation. c) As b), but where the SO index is less than -1 standard deviation. SHA has also been smoothed over 12 months.**

In order to explore the imprint on steric height of the two major modes of Southern Hemisphere climate variability, we compare composites of the SHA in months with a positive and negative Southern Oscillation Index (SOI) and Southern Annular Mode (SAM) index. We apply a 1-year filter to the SOI and choose months in which the SOI is greater/less than 1/-1 standard deviations to represent positive/negative SOI years, then take the average of the 1-year filtered SHA within these months for the composite (Fig. 6a). For the SAM index, we apply an 18-month filter and consider months in which the SAM is greater



than 2 x the standard deviation as positive and below 0 to be negative (Fig. 7a). This provides a workaround to the problem of the SAM index being mostly positive and increasing over the study period.

The SOI composites show a zonally variable response to the SO (Fig. 6b, c). The key feature is the dipole between the South Pacific and wAP exhibiting a positive SHA in the South Pacific and negative in the wAP during negative SOI phases. The signal in the Ross Sea is coherent with the wAP. The SHA in East Antarctica is less affected by the Southern Oscillation than West Antarctica, suggesting weaker influence from the Pacific. In the Weddell Sea, there is a weakly positive SHA in an otherwise negative area that appears to follow the export pathway for AABW (Morrison et al., 2020). This could suggest changes in dense Weddell Sea water classes in response to SOI fluctuations; however, this is speculative, and changes of this nature are likely to occur over timescales greater than 1 year. We also see weak opposing signals between the Southern Indian ocean and along the coast of East Antarctica, which could indicate another dipole of large-scale pressure changes that has not yet been investigated.

**Figure 7: a) Timeseries of Southern Annular Mode (SAM) Index smoothed over 18 months. Months showing a SAM index greater than 2 standard deviations and less than 0 are highlighted in red. B) Composite plot of steric height anomaly (SHA) within months where the SAM index is greater than 2 standard deviations. C) As b), but where the SAM index is less than 0. SHA has also been smoothed over 18 months.**





The SAM composites also show opposing SHA responses on the wAP during positive and negative phases, with a high SHA during a positive SAM and a low SHA during a negative phase (Fig. 6b, c). Much of the rest of the Southern Ocean, including the Ross Sea, South Pacific and the South Indian Ocean, has an anomalously low steric height during a positive SAM year. The continental shelf around the Bellingshausen, Amundsen and Weddell seas displays a particularly low steric height during

a positive SAM phase, while the remainder of the Weddell and Scotia seas shows a higher steric height. The Cooperation Sea opposes the Weddell and Scotia seas, and mirrors the response closer to the coastline in West Antarctica. These results are particularly relevant considering the increasing trend in the SAM; if the SAM continues its tendency toward a more positive phase, the conditions associated with these steric height patterns may intensify.

## 4 Discussion

### 4.1 Mechanisms of SHA variability

The largest regional-scale fluctuations in steric height occur on interannual timescales, and show a larger magnitude than seasonal changes and long-term trends in most regions. Climatic forcing from the SOI and the SAM may account for much of these changes, particularly in West Antarctica, where meridional wind anomalies are linked to changes in the depth of the Amundsen Sea Low (ASL) (Turner, 2004; Raphael et al., 2016; Turner et al., 2013).

### 4.1.1 The Southern Oscillation Index

The SOI is the difference in surface air pressure between Tahiti and Darwin, Australia. Periods of sustained negative (positive) SOI correspond to anomalously warm (cool) temperatures in the eastern tropical Pacific, known as El Niño (La Niña) events (Turner, 2004). The temperature anomalies in the tropical Pacific propagate to the South Pacific via atmospheric Rossby waves (Karoly, 1989), affecting the depth of the ASL (Turner et al., 2013). During a La Niña event, the ASL deepens and pulls warm

surface water towards West Antarctica (Henley et al., 2015; Holland and Kwok, 2012; Stammerjohn et al., 2008), resulting in a dipole between the South Pacific Ocean and the Bellingshausen-Amundsen seas (Yuan and Martinson, 2001; Kwok and Comiso, 2002). We see this pattern clearly in the composite plot of SHA for positive SOI years, when high SHA in the Bellingshausen-Amundsen seas reflects lower water column density associated with increased freshwater input from sea ice melt (Kwok and Comiso, 2002). The dipole in West Antarctica that we identified in the time series of SHA in the ice-free

ocean can also be explained (Fig. 5a), switching polarity as the SOI moves from negative to positive in 2008.

### 4.1.2 The Southern Annular Mode

SAM is the primary mode of climate variability in the Southern Ocean and describes the variability in the zonally symmetric pressure anomaly in Antarctica relative to the mid latitudes. A positive SAM index denotes negative Antarctic sea level pressure anomalies, and is associated with an intensification and contraction of surface winds towards the pole. While this

results in a negative SHA anomaly over much of the Southern Ocean (Fig. 7b), the anomaly in the Weddell Sea and on the



western edge of the wAP is positive. In the Weddell Sea, positive SAM is associated with an increased cyclonic wind stress curl. This causes a freshening of the deep waters of the Weddell Sea as lower-salinity surface water reaches greater depths, and could explain the increased SHA signal here (Gordon et al., 2020). Positive SAM is also associated with warmer winds and less sea ice on the wAP (Stammerjohn et al., 2008), suggesting that the positive SHA anomalies here may also be due to

freshening. Concurrently, cold, southerly winds increase the sea ice in the Amundsen-Ross seas (Lefebvre et al, 2004), reducing the SHA. The wAP has recently undergone seasonal warming trends, accompanied by a reduction in sea ice, that are strongly linked to tropical Pacific sea surface temperature anomalies (Ding and Steig, 2013). The increases in SHA in the wAP in both the ice-free ocean and SIZ (Fig. 5a, b) indicate that the ocean is likely to be warming and freshening in this region, a pattern that we would expect with an increasingly positive SAM.

The results shown in Fig. 6 and 7 encapsulate the instantaneous responses of the SHA to the SAM and SO; we do not consider lagged effects. For instance, Meredith and Hogg (2006) found that a positive SAM increased Southern Ocean eddy activity with a lag of 2-3 years, resulting in increased heat flux towards the pole. Our analysis suggests that negative SAM years correspond with elevated steric height, but since the time between positive and negative SAM events (as we define them) is roughly 2-3 years (Fig. 7b), it could be that this signal is influenced by a lagged response of the deeper ocean to a positive

SAM. We also do not provide a seasonal breakdown of the SHA response to SAM due to lack of data; however, isolating warm months may provide a clearer picture of the response induced by SAM-related surface winds, as the modulating effect of the winter sea-ice will be absent (Naveira-Garabato et al., 2019).

### 4.1.3     Other mechanisms

Recent work has shown that in-phase and out-of-phase interactions between the SAM and SOI can have an amplified impact

on sea ice concentration and distribution, resulting in dipoles driven largely by thermodynamic processes (Wang et al., 2023). The mechanisms driving changes in ocean heat and freshwater content are thought to result from isopycnal heave due to changes in wind stress (Wang et al, 2021), which also elicit changes in sea ice drift (Holland and Kwok, 2012), sea ice concentration and ice shelf melt (Cai et al., 2023). Variations in the strength and frequency of cyclones over the Bellingshausen, Amundsen and Ross seas on seasonal-to-interdecadal timescales also drive temperature anomalies and changes in sea ice drift

(Fogt et al., 2012).

Our results illustrate the relationship between SHA and the SOI/SAM over periods of 12/18 months respectively, but responses in SST and sea ice to the SOI and SAM can show periodicity of up to 15 years (Ejaz et al., 2022). There are other interannual tendencies in SHA that are not simply explained by the two major climate modes. For instance, the recent SHA increases in the Australian-Antarctic Basin appear to be unrelated, and may have been driven by longer-term climatic change (Shimada et

al., 2012; Menezes et al., 2017).



### 4.2 Novelty and comparison against other studies

The novelty of our study lies in the application of the SHA estimation method to the Southern Ocean, and in the extensive spatio-temporal coverage of the resulting dataset. To the authors' knowledge, the                         SHA data is the first to comprehensively capture changes in steric height, and thus in density, across the entire Southern Ocean at regular

intervals for an extended period of time. This facilitates identification of spatio-temporal fluctuations and patterns, such as the dipole in the B-A Basin (Results), and reveals regional responses to long-term climate variability. The SHA data has the potential to support various oceanographic applications, which we discuss in the next section.

As few other studies have attempted to compute the steric height of the Southern Ocean, opportunities for direct comparison are limited. Rye et al. (2014) used model output to calculate the steric contribution to sea level changes around the Antarctic

shelf from 1992-2007, and found an increasing steric height around the Antarctic shelf due to increased freshwater input from the continent. Restricting the time period of our data to 2002-2007 exhibits a similar signal (not shown), providing some confidence that the SHA signal is capturing the increasing steric height due to freshwater input from the continent during that period. However, Rye et al. (2014) concluded that steric height accounts for the majority of SSH increase over the studied time period, while we find barystatic height the dominant contributor overall.

It is more difficult still to compare against studies assessing changes to Southern Ocean density using hydrographic observations. Our results reveal stark regional disparities in steric height, suggesting point measurements or transects are unlikely to capture the full picture encapsulated in the present findings. The agreement between the SHA signal and documented responses of the Southern Ocean to the SOI and SAM climate modes, discussed in the previous sub-section, provides confidence in the robustness of our method over long time and large space scales. Further, SHA captures changes

happening in the sub-surface water column, and can provide greater insight into the changes' driving factors than studies that consider only surface signals such as SST or sea ice (e.g. Stammerjohn et al., 2008; Holland and Kwok, 2012; Ejaz et al., 2022; Kwok and Comiso, 2002). The coverage of our results also sheds light on lesser-studied areas, for instance, the Southern Indian and Co-operation seas, which also exhibit a distinct response to climate variability and likely contribute to the broader picture of circumpolar changes across the Southern Ocean.

### 4.3 Future applications

Coupled with supplementary data from in situ observations, the SHA data will facilitate the calculation of freshwater budgets for major regions and circulation systems of the Southern Ocean. Lin et al. (2023) use a similar method involving DOT and GRACE to compute the freshwater content of the Beaufort Gyre, using known values of the salinity and density of the region in question. Our SHA data will similarly provide insights into the Ross and Weddell gyres, the freshwater content of which

influences key processes like Weddell Sea Bottom Water formation (Gordon et al., 2020; Meredith et al., 2011). The gyres span areas sufficiently far from the Antarctic coastline that this study would be feasible without needing to address the limitations in the satellite data we describe herein.



Theoretically, steric height computed in this way can be indicative of changes in the stratification of the water column, since a net density increase can indicate convection-favourable conditions, and a decrease a more stratified regime (Gelderloos et al., 2013). This could reveal valuable insights into changes in the structure of the water column and associated water mass transformations. For instance, open-ocean polynyas may develop where suitable circulation conditions over a number of years lead to the build up of sub-surface heat, before intense mixing of the warm waters to the surface (Cheon and Gordon, 2019; Dufour et al., 2017). Tracking stratification changes via the SHA is possible in regions of known polynya development, such as the Weddell and Cooperation seas, and could help predict if a polynya is likely to occur and for how long it might persist. Open-ocean polynyas, such as Maud Rise polynyas in the Weddell Sea, occur sufficiently far away from the coast that the current dataset would suffice, and improvements to coastal gravimetry would not he required. Coastal polynyas may require improvements to the gravimetry dataset in order to study reliably, year-round.

Following the same physical rationale, another application for the SHA data could be to detect changes in deep water production in key regions around Antarctica, such as the Weddell and Ross Seas, Prydz Bay and the Adelie Coastline (Morrison et al., 2020). Antarctic Bottom Water (AABW) is produced where shelf waters become dense and sink to the deep ocean, manifesting as a decreased steric height. Many studies describe the recent contraction and freshening of AABW, and possible slowdown of its formation (Zhou et al., 2023; Li et al., 2023; Gunn et al., 2023; Purkey and Johnson, 2013; van Wijk and Rintoul, 2014). However, these largely rely on model results and localised observations. The SHA data could provide a foundational resource with which to integrate these studies, improve upon future model development, and fill a key gap in the Southern Ocean Observing System (Silvano et al., 2023).

## 5 Conclusions

We have shown that the steric height anomaly of the Southern Ocean south of 50°S can be calculated by removing the barystatic height signal obtained from GRACE from the total SSH derived from altimetry. The steric height anomaly follows the expected seasonal cycle in the ice-free ocean and varies zonally within the seasonal ice zone. There is no overall trend in SHA, but interannual regional oscillations dominate. The strongest signals in the SHA timeseries appear to be driven by atmospheric teleconnections associated with the SOI and SAM, with particularly large fluctuations on the wAP and in the Bellingshausen and Amundsen seas.

This method works well in the ice-free ocean and seasonal ice zone, but there is not enough data yet to comprehensively validate it closer to the Antarctic continent, in the permanent ice zone and on the continental shelves. Here, the reliability of the method varies zonally and seasonally due the GRACE leakage error and anti-aliasing. This results in increased uncertainty close to large ice shelves, rapidly melting/freezing ice sheets and glaciers, and areas of the ocean for which (e.g., wind-driven) barotropic processes drive rapid changes in the spatial distribution of ocean mass.

We suggest that addressing the uncertainty in GRACE could permit further analysis of the SHA on the Antarctic continental shelf. Other variants of GRACE mascon data, available from GFZ or JPL, where different leakage algorithms have been

applied, may improve the oceanic signal on seasonal timescales. There are also regular updates to the anti-aliasing model which, once applied to the GRACE mascon products, may improve the quality of the regional mass signals by addressing small-scale features; for instance, the impact of cavities in ice shelves, potentially resulting in a large error at certain Antarctic locations. Finally, gravimetry data that are more heavily supported by models and altimetry are emerging, which may provide a more reliable source of barystatic height.

## 6    Data availability

GRACE Mascon data was downloaded from https://www2.csr.utexas.edu/grace. The marine mammal data were collected and made freely available by the International MEOP Consortium and the national programs that contribute to it. (http://www.meop.net). Argo float data were collected and made freely available by the International Argo Program and the national programs that contribute to it. (https://argo.ucsd.edu, https://www.ocean-ops.org). The Argo Program is part of the Global Ocean Observing System. Sea ice concentration data are provided by NSIDC and have been downloaded from https://polarwatch.noaa.gov/catalog/ice-sq-sh-nsidc-cdr-v4/preview/. The Southern Oscillation Index is provided by NOAA and was downloaded from https://www.ncei.noaa.gov/access/monitoring/enso/soi#calculation-of-soi. The Southern Annular Mode Index is provided by British Antarctic Survey and was downloaded from https://legacy.bas.ac.uk/met/gjma/sam.html.

## 7    Code availability

The code to reproduce all figures is available in the repository at https://github.com/eejco/stericheight_2023.

## 8    Author contributions

J. Cocks performed the data analysis and compiled the manuscript. A. Silvano, A. Naveira Garabato, A. Marzocchi and A. Hogg provided supervision and technical input. O. Dragomir provided altimetry data and support working with this. N. Schifano contributed to development of methods and initial ideas.

## 9    Competing interests

The authors declare that they have no conflict of interest.

## 10    Acknowledgements

Funding for this research was provided by NERC through a SENSE CDT studentship (NE/T00939X/1).



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

## 12   Appendix A: Further validation of SHA

### 12.1   Validation of SHA on the continental shelf

We isolate the SHA signal on the continental shelf by restricting the area to south of 60°S and shallower than -1000 m (Fig. S1). From June to February, the SHA approximately mirrors the sea ice curve, reducing and indicating denser water during more icy periods, and increasing during warmer periods with higher surface freshwater fluxes. However, the increase from March to May is unexpected and difficult to explain physically. The March-May increase is identical to the signal from DOT alone, since GRACE only shows a small increase of <1 cm during these months (not shown). This suggests a large increase of
3 cm in SHA arising from a density decrease on the shelf from February to May, a period of cooling and formation of sea ice, which seems unlikely. Such pattern is not seen in the gridded GPH data (not shown); however, due to the nature of the spot-sampling of the profiles from the Argo floats and seals, it is possible, though unlikely, that the March-May increase has been aliased in the GPH dataset.

We look more closely at the Ross Sea to understand this unexpected signal. A handful of Argo floats provide multi-year
readings in this area, from which a seasonal cycle can be derived. Here, we use GPH from individual floats rather than the gridded GPHA, due to spatial and temporal sparsity. A single float (5904152) is selected for its good temporal coverage and relatively small distance travelled, so we can be sure that temporal changes are not affected by the location of the float. The seasonal cycle of GPH calculated from this float exhibits the expected decrease between March and May, and not the increase seen in the SHA data (Fig. S2). The seasonal cycle from the single float agrees more closely with the observed seasonal cycle




from multiple autonomous profilers from the Ross Sea continental shelf, which show minima in temperature and salinity around February and March (Porter et al., 2019), providing confidence that our selected float is not an exceptional case.

While the steric signal from March to May appears erroneous, the DOT signal from which it is derived is documented and understood. The peak in DOTA in May is caused by high westward wind stress driving water towards the coast, particularly from the Ross and Weddell gyres, resulting in an increase in the SSH (Armitage et al. 2018; Dotto et al., 2018). This kind of

mass transport should increase the barystatic height on the coast; however, no such signal exists in the GRACE data (Fig. S3). This would suggest that the change in SSH is arising solely from an increase in steric height, where cold and salty coastal water is replaced by warmer, and perhaps fresher, water from lower latitudes, leading to the May spike in SHA. While this effect may be contributing, it is unlikely to be the sole cause, and we would expect a strong barystatic signal to account for most of the seasonal increase in DOT.

GRACE is lacking in validation studies close to the coast and in polar regions. Comparison of the GRACE signal against ocean bottom pressure on the Antarctic continental shelf has shown good agreement in some locations; however, the shape of the seasonal cycle of mass changes rapidly with distance as the coastline is approached (Hayakawa et al., 2012), such that a positive validation result at one location is unlikely to hold for all areas of the coastline. It is also known that the leakage error (resulting from gridding from spherical coordinates) increases where there are coastal ice sheets, glaciers, rapidly changing hydrology

and steeply sloping bathymetry (Chambers, 2006), all of which abound in the Antarctic continental shelf region.

Another explanation for the apparent lack of a barystatic signal in the Ross Sea may reside in the specifics of GRACE processing. Barotropic ocean mass movements, like those described by Armitage et al. (2018) and Dotto et al. (2018), can arise on short timescales of less than a month. GRACE accumulates observations over a month and is corrected against an anti-aliasing model from which high-frequency variations have been removed (Dobslaw et al., 2017; Save et al., 2016). It is

possible that high-frequency barotropic events, such as the rapid onshore mass transport occurring during March-May in the Ross Sea, are being omitted. We provide a more detailed explanation of the GRACE uncertainty in the context of this study in Appendix B.

## 12.2 Sources and implications of uncertainty

The SHA computed from altimetry and GRACE shows good agreement with observed GPHA north of about 65°S. However,

at higher latitudes, closer to the Antarctic continent and within the seasonal ice zone, validation is more difficult due to sparsity of observations. Using single-float comparisons we find seasonal disparity, with months from June to January showing good correlation but February to May exhibiting large disagreements (not shown). This could be due to regional or seasonal error or uncertainty in the DOT/GRACE data; differences in scale and sampling frequency between the satellite and profile data; or other effects, because the altimetry-gravimetry method does not strictly measure steric height, and only provides an

approximation.

Known DOT and GRACE uncertainties are described in their respective literature (Data and Methods). Uncertainty in altimetry data is impacted primarily by the sea ice cycle, potentially resulting in seasonally and regionally varying errors (Holland,



2014). The smoothing of the DOT dataset, and conversion of GRACE to a grid from spherical coordinates, are likely to alias smaller-scale features that could be significant even when averaging over larger areas, particularly close to the coast. GRACE
uncertainty is higher nearer areas of high mass or rapidly changing mass, such as the Antarctic continent and ice shelves, but may perform well close to the continent in some cases where ice shelves and glacial activity do not distort the ocean mass signal, for instance, in Lützow-Holm Bay, East Antarctica (Hayakawa et al., 2012; see Appendix B).

Based on the results of this validation, we choose to focus our analysis on the open Southern Ocean, avoiding uncertain areas that are close to the coast, near to large ice shelves or glaciers or subject to high-frequency mass transports, such as the coastal
Ross and Weddell seas. Despite not having full coverage in our validation data, we include analysis within the seasonal ice zone, as it is sufficiently far from the continental shelf to be unaffected by the uncertainty in that area. We further minimise errors by considering changes only over wide time and space scales, such that seasonal or local errors will be smoothed out. We provide suggestions for progressing this method to better account for smaller and more coastal regions in Appendix B.

## 13   Appendix B: Notes on GRACE

### 13.1   Introduction

GRACE refers to the satellite mission that uses twin satellites to observe the Earth's changing gravity field, from which measurement of water mass redistribution both on land and in the oceans may be obtained. GRACE records data using spherical harmonics, a mathematical framework that employs orthogonal functions to encode the spatial changes in the gravitational anomaly across the surface of the globe. It is possible to work directly in spherical harmonics; however, for the purpose of this
work it is more practical to use a gridded solution that has been derived from the spherical harmonics.

There are three centres that perform this derivation to create a gridded product: the Jet Propulsion Laboratory (JPL), the University of Texas Centre for Space Research (CSR), and the German Research Centre for Geosciences (GFZ). Each centre uses a different method to map the spherical harmonics onto a grid, and regularly updates their products in line with new research and technological improvements. The gridded products contain global mass concentrations ('mascons'), which are
represented over the ocean as the liquid water equivalent change in sea surface height, in metres.

### 13.2   Leakage error

We use the CSR GRACE/GRACE-FO RL06 Mascon Solutions (Save et al., 2016; Save, 2020). These provide the liquid water equivalent height anomaly relative to a baseline period of 2004- 2009. The mascons are provided on a grid of about 120 km-wide cells, but the resolution of the underlying data is limited by the way in which it is originally captured using spherical
harmonics, and has an estimated resolution of 300 km (Save et al. 2016). The data have not been smoothed, but the documentation advises a minimum study area size of 200 km.

GRACE gravimetry data is subject to significant uncertainty from leakage error and coarse resolution (Chen et al., 2019; Karimi et al., 2022). Both are due to the way in which the gravitational anomalies are initially recorded as spherical harmonics,



the degree which determines the resolution, as well as the high altitude of the GRACE platforms and resulting wide footprint.
This means that when converted to a gridded product, the stronger gravitational signal from the land 'leaks' into the ocean grid
cells, since there is no step change in how they are originally captured. The leakage error is increased where there are coastal
ice sheets or glaciers, rapidly changing hydrology including heavy precipitation or freshwater fluxes (Chambers, 2006), and
steeply sloping bathymetry, all of which are commonplace in the Southern Ocean. The effective resolution and subsequently
the uncertainty of barystatic height in areas close to the Antarctic continent is therefore likely to be far greater than what has
been documented for the global ocean or better understood regions (e.g., Kuo et al., 2008).

We do not smooth our data. Other studies report smoothing by between 250 km to 500 km before using the data, including a
coastal buffer of 300 km (Johnson and Chambers, 2013) or up to 1000 km for earlier versions of the data (Chambers, 2006).

### 13.3 Comparison against ocean bottom pressure

GRACE ocean mass measurements correlate well with ocean bottom pressure recorders (Save et al., 2016) and altimetry-Argo
data (Chet et al. 2020, 2022; Purkey et al., 2014) on global scales. However, there are few studies in polar or coastal regions,
which are known to suffer from increased leakage errors.

GRACE can be validated against Bottom Pressure Recorder (BPR) data, which is analogous to ocean mass at a particular
location. We use the JARE BPR in Lützow-Holm Bay, East Antarctica (Hayakawa et al., 2012). Hayakawa et al. (2012)
compare the BPR data against an earlier version of GRACE. We repeat the comparison using the GRACE RL06 Mascons, and
find good agreement with the JARE BPR, even when using unsmoothed data (Fig. S4). The seasonal cycle using only data
from 2003-2009 has the same shape, showing a minimum in May and July and peaking in austral summer.

Hayakawa et al. (2012) also consider a second SSH measurement at the Syowa Tide Gauge, roughly 2° to the South and East
of JARE BPR. The OBP inferred from this shows an opposite seasonal cycle to the JARE BPR due to annual variability in
upwelling caused by Ekman divergence. This demonstrates that a) large differences in seasonal cycle can occur close to the
coast, within distances smaller than 300 km that are thought to be the spatial resolution of GRACE, and b) different
mechanisms dominate during different times of the year for different locations, which we know from comparing to Dotto et
al. (2018).

### 13.4 Anti-aliasing model AOD1B

Due to sampling limitations, GRACE must accumulate observations over 7-30 days to achieve a reasonable spatial resolution
(Dobslaw et al., 2017). Gravitational variations within the accumulation period must be corrected for when the mean is taken,
for instance, tidal and non-tidal variations such as heavy precipitation and surface winds driving barotropic ocean mass
transport have time periods shorter than 30 days. If these sub-monthly variations are aliased into the GRACE signal, this results
in meridional striations.

To correct for these variations, Stokes coefficients to account for mass changes due to atmospheric pressure and dynamic ocean
effects have been computed in AOD1B RL06 and used to anti-alias the GRACE measurements. The high-frequency variations



are removed during the early processing stages, then the monthly means are re-added to the final GRACE product for oceanographic studies (Save et al., 2016; Dobslaw et al., 2017). The anti-aliasing corrections performed using the AOD1B model are referred to as the GAD corrections.

The mean seasonal cycle of the uncorrected GRACE mascons (GRACE-GAD, Fig. S6) closely resembles the global mean (Johnson and Chambers, 2013), since it takes only a few days for mass change from water mass fluxes to be distributed globally (Lorbacher et al., 2012). The mean GAD seasonal cycle has a smaller amplitude and two minima in May and October, reflecting the more localized and short-timescale changes. The multi-decadal rise in barystatic height is predominantly from the uncorrected GRACE mascons; however, the GAD also shows an increase from 2014 to 2018, which may reflect more local barotropic responses to changes in winds, or gravitational effects from loss of land ice.

The SHA signal during the austral autumn in the Ross Sea behaves unexpectedly, due to a rapidly increasing DOT that is not reflected in the barystatic height (Fig. S3). The cause of the increasing DOT is thought to be wind-driven mass water movement towards the coast, a barotropic effect with a relatively short timescale. This type of mass movement can be aliased in the monthly GRACE mascons, but should be captured by GAD and corrected in the final product.

The GRACE data without GAD shows a near-zero SHA in the Ross Sea in autumn, but a positive anomaly in the Weddell Sea 740 (Fig. S7). The GAD product alone has a negative anomaly for the barystatic height at the coast, resulting in a negative barystatic height from GRACE once the GAD has been restored. Due to the strongly positive SSH anomaly in autumn in the Ross Sea (Fig. S1), which is explained by barotropic mass transport, we would expect to see this represented either in the GRACE monthly average or the GAD product. This could be a problem with the product and needs further investigation.

### 13.5 Suggestions for further work

Additional studies could be performed using the GRACE OBP data from JPL or GFZ. There is a GFZ dataset available in which spatial leakage has been reduced to permit analysis within coastal zones (Dobslaw et al., 2020). The JPL dataset has had a coastal resolution filter applied (Weise et al., 2018). The raw gravimetry data and the AOD model are the same for the three centres, however the processing is performed differently for all three, including the method with which the anti-aliasing is applied. This may result in a large difference between data from the different centres in regions where there is enhanced 750 uncertainty, e.g., near ice-shelves or for sub-monthly processes.

The GRACE anti-aliasing model A0D1B has recently been updated to a 7th version (RL07, Shihora et al., 2022). RL07 includes cavities in Antarctic ice shelves, which were previously omitted and thought to have a significant impact on local ocean mass. Using a GRACE product that is processed with AOD1B RL07, once such a product becomes available, could improve the validity of the SHA on the shelf in the Ross and Weddell seas.

Alternatively, Dobslaw et al. (2017) suggest adding the GAB, instead of the GAD, fields back to the GRACE gravity fields. GAD is the monthly mean of the gravitational fields from both atmospheric and oceanic sources, whereas GAB contains only the ocean component. Using GAB means that the atmospheric contribution to gravity anomaly, including inverse barometer corrections, would be omitted. In this study, this could be beneficial since the DOT has already been IB-corrected.





Finally, gravity models enhanced with altimetry data are becoming available (e.g., Zhu et al. 2022), which may offer improved

resolution and reduced coastal leakage. Replacing the GRACE Mascons with data from a gravity model, or satellite data from

a future gravity mission, may improve results in the shelf seas and close to large ice shelves.