# Peer review of "Satellite-derived steric height in the Southern Ocean: Trends, variability, and climate drivers"

_EGUsphere, 2023_

## Author Response (AR1)

Reviewer 1:

We thank the reviewer for their detailed consideration of our submission and for making useful suggestions which we have now incorporated in the updated manuscript.

Major comment:

| Reviewer comment | Author's response | Changes to manuscript |
|---|---|---|
| The title appears broadly misleading judging by the content of results. Heat and freshwater changes are not estimated, so their contribution to steric height changes is discussed only in a speculative way. This is fine, but the title should reflect core results, not discussion. | We agree with this. | The title has been changed. |
| A more quantitative estimate of errors would be useful. You are merging information from two very different products. Altimetry has a better spatial resolution; GRACE has some problem of "leakage" (contamination by continental surfaces); temporal resolution between the two products might also differ. What would be the combined uncertainty on steric height estimates and is this expected to be sufficient to capture different scales of variability? | We agree with this in theory, however, the uncertainty on GRACE in this area is poorly understood. Global and regional estimates of GRACE uncertainty are available in the literature but it would not be appropriate to use those here, as it is thought to vary dramatically depending on bathymetry, proximity to land, presence of ice shelves and short-term barotropic mass movement. | We have added a section (Section 4.2) in the discussion which addresses the uncertainty in more detail. We have also added a paragraph (beginning Line 209) discussing the margin of error in our Kerguelen Island validation and comparing against the seasonal and interannual variability. We conclude that a comprehensive calculation of uncertainty is out of scope for this study. |
| There is a problem with the smoothing applied on temporal curves in Fig. 4, 6 and 7. Judging by eye, the rolling mean is not centered, so it appears lagged by half the smoothing window (i.e. 6 months). This need to be checked, and I apologize if my eye is inaccurate. | You're correct. | This has been corrected in Figures 3, 6 and 7 (Figure 4 has been removed). |
| The potential of the presented dataset appears under-used in the study. Why not add a map of steric height anomaly RMS? EOF of steric height changes would also be useful. A regression of steric height changes to SAM and SOI would help demonstrate a potential statistical link, together with some quantification of significance. At the | Thank you for these suggestions. We do not show regressions of the steric height against the climate indices as the composite plots are more illustrative of the response. | We have added a section (Section 3.2) in the results showing the variation and results of an EOF analysis. We discuss these further in the Section 4 and tie into the SAM and SOI analysis and the existing discussion. |

| | | |
|---|---|---|
| moment, we struggle to see any clear novelty in the analysis, despite the obvious value of the dataset and potential of the method. | | |

Minor comments:

| Reviewer comment | Author's response | Changes to manuscript |
|---|---|---|
| l. 10: "The Southern Ocean *circulation* plays …"

l. 27: Reference Haumann et al 2023 absent from reference list. Check all references.

l. 29: "The South Atlantic [..] Oceans *waters* …"

l. 31: "extreme and unpredictable", vague and potentially misleading. The weather might become more extreme, not sure about the climate. Not sure about the future of climate predictability either. | Noted, thank you for highlighting these. | These have been corrected. |
| l. 172: I do not find the agreement between SHA and GHPA particularly striking in Fig. 3e. Fig. 3b seems more convincing. Can you be more specific/quantitative? | We acknowledge more quantitative comparison was needed. | We have re-worded Section 3.1 to include quantitative comparison and clarify the significance of Figure 3f (previously 3e). |
| We politely suggest the authors to read and incorporate the citation Kolbe et al. 2023 in their work (a paper that went unnoticed as so many other studies from the Covid era), as they will find much information about steric height variability in the Southern Indian Ocean during a period overlapping their period of interest, that may be compared with their results. Notably, the relative importance of heat and freshwater changes is discussed in detail. | Thank you for the suggestion, this paper has been an excellent resource for the revisions. | We have plotted the SHA trend (Figure 4a) as it is useful for validation via comparison against the Kolbe et al study (Line 468). We also refer to the latter in our Discussion when interpreting the trend (Line 378) and drawing links to interannual climate modes. |

| | | |
|---|---|---|
| •There is too much information in the Appendix. Appendix A is useful to interpret the main results. Appendix B could be shortened and put back into the main text. The validation of GRACE against ocean bottom pressure is useful and could be part of the main text. Fig. S4 should also be in the main manuscript. Section 13.5 has nothing to do in an Appendix. | Noted. | We have removed Appendix B and included a section (Section 4.2) in the results assessing the uncertainty of GRACE in context of the present study. The validation of GRACE was simply a repeat of what had previously been performed by Hayakawa et al. (2012) so we have removed this and refer to that study instead. |

References:

Hayakawa, H., Shibuya, K., Aoyama, Y., Nogi, Y., & Doi, K. (2012). Ocean bottom pressure variability in the Antarctic Divergence Zone off Lützow-Holm Bay, East Antarctica. Deep Sea Research Part I: Oceanographic Research Papers, 60, 22–31.

Kolbe, Marlen, Fabien Roquet, Etienne Pauthenet, and David Nerini. "Impact of Thermohaline Variability on Sea Level Changes in the Southern Ocean." Journal of Geophysical Research: Oceans 126, no. 9 (2021): e2021JC017381. https://doi.org/10.1029/2021JC017381.

Reviewer comment 2:

We thank the reviewer for their detailed consideration of our submission and for making useful suggestions which we have now incorporated in the updated manuscript.

Major Comments.

| Reviewer comment | Author's response | Changes to manuscript |
|---|---|---|
| I agree with Anonymous Referee #1 that title is not appropriate. The manuscript does not assess the Southern Ocean heat and freshwater changes, it does estimate the steric height changes. | We agree with this. | The title has been changed. |
| I suggest the authors revise how information is written in the manuscript. It is important that clear and concise language that correctly describes the applied data methods is used. For example, but | Noted, thank you for highlighting this. | We have re-written Section 2.4 to clarify the procedure. We have also proof-read the Methods |

| | | |
|---|---|---|
| not limited to, section 2.4 explains how in-situ data are used to calculate geopotential height anomaly. It is unclear what is the value of the statement " in which there are multiple pressure level" when the depth criteria are used to determine what profiles are used (Pmax > 500 dbar and Pmin<25 dbar and quality flag 1). Rather than beginning the sentence with "The maximum depth of 500 dbar…..", perhaps it is better to begin with " We calculate GHA relative to 500 dbar, where 500 dbar is sufficiently deep…… | | and Results and re-written any areas where extra clarity could be needed. |
| Another example, in figure 3 b the correlation coefficient is mostly greater than 0.25 but there are region where the correlation coefficient is less than 0 indicating strong negative correlation, and regions of 0 correlation coefficient (no relationship between SHA and GHPA).

The white color in figure 3b may not actually show regions of no relationship but rather region with less than 6 in situ profiles. Here it is important to use a mask for regions where the Pearson correlation coefficient was not calculated.

Similarly, in 3c the color bar axis should have a minimum value of 6 and a mask applied to regions with less that 6 profiles. While these are rather small correction, the accurate display of data is important.

Finally, what is the value of figure 4d and 4e. A reader may ask why you show a region with anomalous in situ observations. Also, for 4d there appear to be data that in water depth less than 500 m. | The purpose of Figure 3f (previously 3e) is to demonstrate the agreement between the satellite and in-situ data in a region where we have temporally abundant in-situ profiles. The correlations we derive on large scales (i.e. whole southern ocean, Figure 3e vary in quality between grid-cells based on the temporal availability of in-situ profiles. For example, some grid squares have only a few profiles collected in summer, meaning that the correlation shown may not capture the year-round relationship. Thus 3f is intended to show a higher quality, temporally-comprehernsive comparison.

It was previously unclear that we were excluding grid cells with less than 6 months' worth of **gridded monthly GPHA data**. This is a different quantity to the total number of profiles in a given cell as shown by Figure 3d (previously 3c), i.e. there could be many profiles within a single month. We have explained this difference in the text and thus have not modified the scale on Figure 3d or added a mask. | We appreciate the detailed suggestions and have revised Section 3.1 with the following changes:

Addition of comparisons of seasonal cycle of SHA, GPHA (these were previously shown in a different figure which has been removed as per another review comment) and sea ice concentration to support the validation analysis.

Removal of Figures 3a and 3d.

Coarsened grid cells for the correlation analysis.

Clarified the gridding procedure (Line 188).

We now use 36, rather than 6, months of data as the cutoff point which has been scaled in keeping with the coarser grid cells.

Added a quantitative analysis of Figure 3e (correlation plot, previously 3b) (Line 195).

Added masked region on Figure 3e (previously 3b) where there were fewer than 36 months' worth of data.

Clarified the significance of the Kerguelen Island study (i.e. Figure 3f (previously 3e)) (Line 199). |

| | | |
|---|---|---|
| Lines 176 to 180 need to be revised. How you validate the SHA is the sea-ice zone is important. What is meant by " open Southern Ocean"? 60oS to the seasonal sea-ice zone? | Thank you for highlighting this. | These lines have been removed, along with any reference to the 'open Southern Ocean' and replaced with more specific regions. |
| While the interannual SHA variability shows a potentially interesting signal, it is somewhat complicated, and the analysis undertaken needs further investigation and refinement. This more detailed consideration of the causes of the variability would enable the authors to reach much more conclusive findings.

I would suggest that the authors consider only the SHA and the relationship with the large-scale SAM and ENSO forcing for this current manuscript and continue to investigate the interannual and seasonal signals for inclusion in another manuscript. | Thank you for the suggestion, this has helped a lot with the revisions. | We've removed results concerning the SHA time series and seasonality, and instead focussed on additional statistical analysis which ties in with the SOI and ENSO analysis. |

---

## Author Response (AR2)

Letter to the Editor: 2nd Resubmission of Manuscript

Dear Bernadette,

Many thanks for your continued support towards our manuscript submission. Please see below our responses to the second reviewer's comments and the corresponding changes we've made to the manuscript.

Best regards,

Jennifer Cocks
* * *
We thank the reviewer for taking the time to consider our manuscript and for their positive and constructive response.

Major comment:

| Reviewer comment | Author's response | Changes to manuscript |
|---|---|---|
| I think introduction would benefit from a bit more information about why computing steric height from satellites is difficult in the Southern Ocean. Is it just the Steric and barystatic height that is challenging? It feels like the thing that makes this study novel, and I feel like this isn't highlighted enough, especially in the intro. This paragraph seems like a good place to reference to figure 2. | We agree – thank you for picking this up. | We developed a paragraph in the introduction to give greater detail on previous work on this subject and the difficulties of transferring it to the Southern Ocean, please see lines 69-81. |
| When calculating geopotential height, I'm concerned about the choice to reference to only 500 dbar. I understand that changes in water mass properties will preferentially materialize in the thermocline, but I would still probably reference to 2000 or 1800 when using Argo. Why did you choose 500? You should have enough profiles in | We chose 500 for the following reasons:

- Many of the profiles come from tagged elephant seals, who rarely dive beyond 500m.
- The Southern Ocean follows an equivalent barotropic circulation model, and the steric | We have added a paragraph into the methods section to explicitly explain this reasoning, please see lines 156-163. We refer to the papers Meijers et al., 2011 and Killworth and Hughes, 2002, who show that the difference between surface GPH change and that at depth is minimal. |

| | | |
|---|---|---|
| the Southern Ocean so that you still have sufficient data even when eliminating shallower profiles. Especially when comparing to steric height anomaly which will represent the entire water column. Also, in the introduction, you motivate this work by discussing changes in deep water which you won't capture by only referencing to 500. Can you either justify your choice or show that it doesn't change your results to use 500 dbar as opposed to 2000. | height changes in the top 500m are highly correlated with the steric height changes at depth (we have performed our own check of this on the Argo floats to check it holds true for our data – please see Figure 1 below).
- We would like to use the same validation procedure both on and off the shelf, so a shallower reference depth is required. | |
| Figure 6: the overall trend is really interesting. You say the trend is predominantly negative but there are large regions where it's positive. I feel like the manuscript is missing some discussion of what possible physical drivers are causing positive and negative steric height trends. I guess I'm surprised that the trend isn't positive everywhere and curious why that would be. | We think the reviewer might mean Figure 4. We agree there should be a clearer discussion of this as it is unexpected! | We have rearranged and expanded the discussion (Section 4.1) – we now begin by discussing the trend, before looking at SAM and SOI. We've added more in-depth discussion around the widespread decreases to SHA relating to the MOC, please see lines 60-68. |
| The seasonal cycle should be removed before calculating the EOF in figure 5. It looks like it dominates the first and second modes. It probably also wants to be removed from SOI and SAM analysis. | Thank you for these suggestions. We have experimented with removing the seasonal cycle from the EOF analysis (please see Figure 2 below) and have decided to retain it for the following reasons:

By including the seasonal cycle, we are able to draw conclusions on the role of the seasonal cycle towards the overall variability, for instance, we were able to show that the seasonal cycle is dominant in the first mode, with a much larger amplitude than interannual changes. Thus, we can conclude that the seasonal cycle has a dominant influence on overall SHA variability, and we can explore the spatial pattern of this in the EOF plot. We think this is informative to show.

Furthermore, removing the seasonal cycle seems to have the effect of removing the first mode, so that Mode 2 becomes Mode 1, and Mode 3 becomes Mode 2 (minus the low-amplitude | We include a line (L277) to summarize our reason to retain the seasonal cycle in the EOF. |

| | seasonal cycle in the time series). Modes 3 and up have a very low Explained Variance Ratio and lack coherent spatial patterns, so there is no benefit to removing Mode 1 (i.e., removing the seasonal cycle).

We also experimented with removing the seasonal cycle from the SOI and SAM composite plots. However, since we're already smoothing the SHA used here, removing the seasonal cycle has a negligible effect (a maximum of 0.1cm, the difference was not visible in the plots). | |
|---|---|---|
| You reference a Kolbe 2024 paper that used GLORYS to do a similar analysis but I can't find the paper in your references. There's a Kolbe 2021 paper though. Please check that all your citations are correct and included in your references. | 2024 was an error – it should have been 2021. Thanks for picking this up. | Reference corrected. |
| Line 463: GLORYS is a reanalysis product that assimilates satellite altimetry and Argo so it isn't surprising that your results are similar here. I think you should expand on this paper in more detail, discuss what they found and how your results add to our understanding beyond what this paper finds. Should probably include some discussion of this in the intro as well (probably all papers in section 4.3). See above note about reference. | | We have elaborated our reference to Kolbe et al. (2021) in Section 4.1 with more information on the mechanisms resulting in the decreasing steric height trend.

We've added a line in Section 4.3 (L498).

We have added a discussion of previous work to the introduction, please see lines 60-68. |

Minor comments:

| Reviewer comment | Author's response | Changes to manuscript |
|---|---|---|
| | | |

| | | |
|---|---|---|
| Sentence starting on line 32: A bit awkward and very long. Split into two. Maybe be a bit more specific about what has changed about deep water and the overturning circulation. | | Line has been modified and broken into two. |
| Line 44: maybe change to, including the marginal sea ice zone? Or including sea ice regions. Maybe just 'sea surface height of the Southern Ocean, and more accurately' | | Changed to 'marginal ice zone'. |
| Section 2.1: Was there a reason you did your own gridding as opposed to using a pre-gridded product that might extend beyond 2018? | The data were gridded by Dragomir (2023), but this wasn't clear in the text. | Clarified that we did not process the DOT data. |
| Line 131: This is a bit awkward and seems repetitive. Considering possible changes in steric height, would your SSHA calculation be more accurate if you computed it from, for example, a 180-day low pass filter rather than the time mean? | We must compute anomalies relative to a time mean since the GRACE data are provided as anomalies against an arbitrary time mean. Computation against anything else is impossible, since there is no reference. | We have shortened this section and removed repetition. |
| Figure 2 is referenced before figure 1. Should probably reference figure 1 somewhere in the intro. | | Figures have been swapped. |
| Section 2.4: You should include how many Argo floats/profiles were used, the years your float records span and maybe something about their distribution in space and time? I think profiles were gridded? That information should be included too. | Agreed. | We've added the total number of profiles (L152) and a supplementary figure to show the distribution of profiles in space and time (see Fig. 5 below). We refer to this in the text at L153.

We also moved the explanation of the gridding from Section 3.1 into Section 2.4. |
| Line 175: follows from above comment, you say you subtracted mean geopotential height from July to June 2018 but you don't have any Argo profiles from 2002. Probably worth including some discussion about when you have a high enough degree of freedom to accurately validate using the Argo dataset. | We discuss this in Section 3.1 and describe the following measures which have been taken to ensure sufficient data are available for validation:

- Coarsening of the grid by a factor of 6
- Excluding grid cells where we have fewer than 36 months' worth of data | Supplementary figure S1 (please see Figure 5 below) added to show temporal and spatial distribution of profiles.

We added a line (L200-202) to clarify that these measures were taken to achieve a good enough degree of freedom for validation. |

| | - Performing local comparison in a location where we have a high density of data | |
|---|---|---|
| Figure 3 caption: you say the red box in panel d outlines the region from which the data in e have been taken but I think you mean the data in f. | Quite right, thank you for spotting this. | Changed to (f). |
| Figure 4 is never referenced – should include somewhere in paragraph near line 235 | Thanks for spotting this. | Added references to Fig. 4a and b. |
| Figure 4 (and elsewhere): maybe consider adding dots or some indication of where validation with Argo data was low/negative or not possible | We show in Figure 3e (yellow area) where validation has not been possible and state in lines 220-226 that we include these areas, and those of negative and low correlation, in the results. We do not mark these areas on the other plots so as not to obscure the results. | We have added a line to the captions of Figures 4-7 to direct the reader to section 3.1 for information on regional variation in data quality. |
| Line 267: Would change this to "temporal component is dominated by an annual fluctuation reflecting steric height changes resulting from the seasonal cycle" | Thank you for the suggestion – nicely worded | Line changed to this. |
| Figure 7: The years identified as negative SAM years don't really seem representative of true negative years, and I don't think enough years are included to consider a reliable composite. Considering the trend, you could just not show a composite of negative SAM years. Alternatively, instead of negative SAM years you could change to weak SAM years. In this case, I would remove the increasing trend from the time series, probably reduce the filter size (12 months max, maybe even less than that) and identify years from there. | We agree that there are very few -SAM months and the few months which are highlighted may not be representative.

We experimented with detrending the SAM signal and reducing the filter to 12 months (please see Figure 3). Since the amplitude of SAM appears to increase over time, this emphasises the latter portion of the time series, and considering the trend, we see a -SAM response which is similar to the +SAM response. | We removed the -SAM plot, and retained only the +SAM. The text in Section 3.3 has been modified to reflect this. |

| | We also experimented with reducing the filter size to 12 months for the non-detrended signal (please see Figure 4). We found a weak signal which seemed to indicate that these were not significant - SAM events, and the algorithm had just found a more random assortment of months. | |
|---|---|---|
| Line 307: this could be better worded. It's not really a problem that the SAM index is positive – it's the signal. | Noted | This has already been changed in response to a previous comment. |
| Section 4.3 I feel like an introduction to these studies should go in the intro. | Agreed | This has already been added in response to a previous comment – please see lines 60-68. |

References:

Killworth, P. D., & Hughes, C. W. (2002). The Antarctic Circumpolar Current as a free equivalent-barotropic jet. Journal of Marine Research, 60(1), 19–45.

Meijers, A. J. S., Bindoff, N. L., & Rintoul, S. R. (2011). Estimating the four-dimensional structure of the Southern Ocean using satellite altimetry. Journal of Atmospheric and Oceanic Technology, 28(4), 548–568.

[Figure]

*Figure 1: Comparison of geopotential height (GPH) referenced to 500 and 1800 dbar from a sub-selection of Argo float data. (left) The geographical locations of the floats in the sample (red, 'x') plotted over the GEBCO bathymetry. (right) The GPH of each profile referenced to 500 dbar plotted against the GPH referenced to 1800 dbar.*

[Figure]

Figure 2: EOF analysis with seasonal cycle removed. Spatial signatures of Modes 1 (a), 2 (b) and 3 (c). The colour scale is arbitrary but consistent across (a), (b) and (c) and centred at zero. The Explained Variance Ratio (EVR) is shown in the title. The sea ice maxima is demarcated in white (solid), and the -1000m isobath in black (dashed). Temporal signatures of Modes 1 (d), 2 (e) and 3 (f). The scale on the y-axis is arbitrary but consistent across (d), (e) and (f) and centred at zero.

[Figure]

*Figure 3: a) Timeseries of SAM index detrended and smoothed over 12 months. Months showing a SAM index greater than 1 standard deviation and less than -1 standard deviation highlighted in pale green. b) Composite plot of SHA within months where the SAM index is greater than 1 standard deviation. c) As b), but where the SAM index is less than -1 standard deviation. SHA has also been smoothed over 12 months.*

[Figure]

*Figure 4: Timeseries of SAM index smoothed over 12 months. Months showing a SAM greater than 2 standard deviations and less than 0 highlighted in pale green. b) Composite plot of SHA within months where the SAM is greater than 2 standard deviations. c) As b), but where the SOI is less than 0. SHA has also been smoothed over 12 months.*

[Figure]

Figure 5: (a) Monthly distribution of Argo and Seals profiles. (b) Yearly distribution of Argo and Seals profiles. (c) Geographical distribution of Argo and Seals profiles.

---

## Author Response (AR3)

Letter to the Editor: 3rd Resubmission of Manuscript

Dear Bernadette,

Thank you for your careful consideration of our revised manuscript and for your comments. We have produced a 3rd revision of the manuscript incorporating your suggestions (detailed below), and an updated reference list in accordance with citation style of Ocean Science. Regarding your query on the re-ordering of co-authors between revisions, the co-authors have agreed that the new order better represents the relative contributions of each author, in particular reflecting contributions throughout the revision process.

Best regards,

Jennifer Cocks
* * *
Minor comments:

| Reviewer comment | Author's response | Changes to manuscript |
|---|---|---|
| Line 58. Suggest change Antarctic waters to Southern Ocean waters, or Antarctic slope waters
Line 131. Suggest you define SHA here. Change to "We compute the steric height anomaly (SHA), by ...."
Line 247 change (d) the to (d) The
Line 265 Change "at the latitudes north of.." to between 50oS and.."
Line 475 What is meant here "….as a function,…" should this be "…as a spherical harmonic function,…" or some other function of?
Line 489 Change South to south and East to east
Line 489 Provide location, latitude and longitude of Syowa Tide Gauge for reader reference.
Line 490 remove "from this"
Line 511 remove (Results) | Thanks for these suggestions. | All have been modified as suggested. |

| | | |
|---|---|---|
| Line 518 provide year of Rye et al study.
Line 525 Remove "discussed in the previous sub-section". If you want to keep the reference to the sub-section, use the section number. | | |
| Line 234. "maxima in 2013". In the figure 3f, it looks like the this maximum is in Isn't this 2012, by 2013 SHA is decreasing?
Line 234. "minima in late 2010". Not sure this is obvious in 3f. The figure suggests minimum in mid-2008 to mid-2009? The minimum in SHA continues through to mid-2010, but not in GPHA which increases from late-2010. | Thank you for these observations: we agree the text could be clearer. | We've changed lines 186-189 to describe more accurately what we see in Figure 3f. |
| Line 237. Figure 3f. Is there any comment on the apparent lead/lag timescale of the 12-month average SHA and GPHA signals? This appear to be evident for maxima anomalies in particular? Is this referred to in Discussion section? | There does appear to be a lag between the SHA and GPHA 2012/2013 maxima however this doesn't seem to be a pattern at any other times so we assume this difference results from errors in the data or the validation procedure. However we agree this should be addressed in the text. | We have added a line (L190-194) to explain the differences between the SHA and GPHA time series, describing how this might arise from either source or due to difference in data scales. |